# Research on coupling risk assessment method of industrialized urban area based on information diffusion and extension grey clustering model

Chen Lv[1,2]*, Xiaolu Wang[3], Sheng Xue[2], Shuang Wang[1]

1 College of Energy Environment and Safety Engineering, China Jiliang University, Hangzhou, China,
2 Key Laboratory of Safety and High-efficiency Coal Mining, the Ministry of Education (Anhui University of Science and Technology), Huainan, China, 3 Department of Automation, Tsinghua University, Beijing, China

* lvchen0707@cjlu.edu.cn

## Abstract

The regional accident risk in industrialized towns is a critical foundation for optimizing urban industrial layouts and dynamically managing regional risks. However, assessing production safety risks in urban areas often encounters challenges such as incomplete information and fuzzy index judgments due to limited small-sample data. This study proposes a dynamic control model for coupling risks in industrialized urban areas, integrating system engineering principles and focusing on two dimensions: the inherent risk of production safety accidents and the vulnerability of regional safety protection systems. For inherent risk analysis, the fuzzy set theory treats probability distribution as a mapping from events to probabilities. A set-valued fuzzy mathematics approach is employed to process single sample points, and an information diffusion model compensates for data deficiencies by appropriately expanding incomplete information. An extension grey clustering vulnerability assessment model is developed for vulnerability analysis, combining extension theory with grey clustering correlation functions. This model calculates the comprehensive correlation degree between the assessment object and vulnerability levels, thereby evaluating the safety system's capacity to withstand accident impacts. Application of the model to an urban district demonstrates that while inherent risks are widespread in industrialized regions, the safety protection systems exhibit minimal vulnerability and robust resilience. These findings align with observed conditions and offer a scientific basis for effectively managing production safety risks in urban areas.

## 0 Introduction

China's economy is fully leveraging the potential of its vast domestic consumer market, with the urbanization rate serving as a primary driver of domestic demand growth in the double circulation development model. Simultaneously, strategically promoting

**Data availability statement:** All relevant data are within the paper and its Supporting Information files.

**Funding:** This research was supported by National Key R&D Program of China (Grant No.2021YFB3301100), Natural Science Foundation of Zhejiang Province (Grant No. LQ20E040005), the State Key Laboratories Program of China (Grant No. JYBSYS2019102.

**Competing interests:** The authors have declared that no competing interests exist.

coordinated urbanization is crucial for China to achieve modernization. According to data from the National Bureau of Statistics website, China is currently experiencing rapid urbanization. The number of permanent urban residents increased from 170 million to 902 million between 1978 and 2022, while the urbanization rate increased from 17.9% to 65.2%. With the rapid pace of urbanization, various factors such as the urban population, construction activities, lifestyle patterns, wealth accumulation, and production processes have become highly concentrated. Particularly in China's context, where different economic entities are undergoing extensive and intensive development on a large scale, diverse forms of economic industrial parks have emerged and gradually integrated into urban areas. Therefore, any major production safety incident occurring within this process could easily result in derivative damages that increase the scope of the disaster while escalating the risks associated with it [1]. As a result, these challenges pose significant implications for ensuring public safety and promoting sustainable development in urban areas across China.

Previously, both domestic and international scholars have conducted extensive research on regional risks associated with individual hazard sources across various fields such as fire protection, construction, chemical industry, and environmental science. Furthermore, they have advanced the understanding of regional risks posed by multiple hazard sources through sophisticated calculation methods that go beyond simple superposition. In 2006, Dong [2] et al. developed a regional risk assessment model based on a linear weighted approach to effectively evaluate urban fire risks. Chen [3] et al. applied the superposition principle to quantitatively assess the overall regional risk status by aggregating the risk assessments of individual hazard sources. In 2009, Cheng [4] et al. introduced a method for calculating the total risk in construction areas using a risk analysis model for single buildings. However, as research on risk assessment theories and methods deepened, it became evident that regional risks could not be simply regarded as the superposition of individual disaster risks. This realization led to the introduction of the concept of coupled regional disaster risks. In 2013, Xue [5] et al. systematically explored the fundamental theories underlying coupled disaster risks, establishing a foundation for further research. In 2015, Lu [6] et al. classified potential risks from seven distinct disaster types using an index-weighted method and developed a coupling model, which, in combination with a risk matrix, provided a comprehensive assessment of multi-disaster risks in urban planning areas. In 2019, Huang and Chen [7,8] et al. proposed that Na-Tech event research should prioritize understanding the mechanisms by which such events unfold and exploring the interactions between equipment vulnerabilities and the multiple disaster triggers under natural disaster conditions. They introduced a comprehensive model for assessing vulnerability to multi-hazard coupling, laying the groundwork for vulnerability studies in chemical industry parks. In 2022, Guo [9] et al. conducted a thorough and systematic classification of urban regional risks, identifying the distinctive characteristics of various risk factors. They developed a multi-agent index system from three perspectives: industrial risk units, urban population-dense areas, and urban public facilities. Utilizing the analytic hierarchy process (AHP), entropy weighting, and multi-correlation analysis, the authors analyzed regional risks

associated with urban public safety from both static and dynamic perspectives. It is crucial to recognize that regional risk constitutes a complex system composed of multiple interacting risk factor [10]. In recent years, the continuous advancement of geographic information technology has brought coupled risk visualization to the forefront of scholarly research. In terms of practical applications, urban emergency management departments in the United States have implemented urban risk assessments and developed risk maps to identify areas with low, medium, and high risk levels. Similarly, France has applied accident risk assessment methods from industrial system projects to construct city-specific risk maps for fire and explosion incidents. This approach enhances data sharing and reuse for monitoring and evaluating risks, while also providing real-time, personalized information for ongoing risk assessments. In 2011, Gai [11] et al. introduced ArcGIS spatial analysis technology to comprehensively assess risk in Beijing by utilizing event chain coupling relationship matrices, thereby achieving effective risk visualization. Additionally, in 2016, Zhao [12] et al. developed a geographic processing workflow model that facilitates severity calculation, vulnerability assessment, and subsequent risk mapping without the need for repetitive and time-consuming steps typically encountered in practical applications. The use of GIS processing technologies has thus enabled the development of a methodological tool specifically designed for regional risk assessments of major urban disasters. In 2018, Hu [13] et al. proposed a process-oriented urban public safety risk assessment model based on the meta-object facility framework, utilizing a unified space-time structure. This method enhances data sharing and reuse for risk monitoring and assessment, enabling real-time personalized information for process-oriented risk evaluations. In 2020, Gallina [14] et al. integrated climate-related hazard data across various spatial and temporal scales, using geographic information system (GIS) maps to achieve multi-hazard coupling risk classification based on exposure and vulnerability factors. However, in urban regional risk calculation, the unclear internal correlation characteristics often lead to missing risk sample information, thereby affecting the accuracy of risk assessment. Moreover, multi-disaster coupling risk analysis remains challenging due to issues such as large data volumes, complex data analysis, inadequate descriptions of disaster consequences, and difficulties in disaster scenario modeling. Consequently, most existing studies primarily focus on qualitative assessments of relative risk levels, while quantitative analyses of comprehensive risks associated with multiple disasters remain insufficient. From the existing research results, statistical analysis methods have demonstrated extensive applicability. Specifically, constructing logistic regression models and establishing disaster data platforms based on experiments and simulations have become typical risk analysis approaches. However, achieving a quantitative assessment of multi-hazard comprehensive risks based on data, models, and computational methods remains challenging. Key challenges include systematically identifying and integrating the critical components and core processes of various types of emergencies, clarifying interaction pathways among different elements within the disaster system, analyzing the structure and fusion techniques for multi-source data, developing an integrated framework for multi-type models, and optimizing the allocation and utilization of computational resources. Furthermore, the impact of uncertainties inherent in current assessment methods and data sources on risk analysis requires further investigation. In summary, numerous issues remain unresolved in urban regional risk assessment. Improving calculation accuracy under conditions of incomplete data and establishing a scientifically rigorous and systematic multi-hazard risk quantification method represent crucial directions for future research.

Information diffusion is a fuzzy mathematical processing method that handles single sample points using set-valued processing [15]. It establishes binary relationships between variables via a diffusion function, which appropriately extends incomplete information to compensate for the limitations caused by small sample sizes. The grey clustering evaluation method, grounded in grey system theory, extracts valuable insights from partial known information, enabling the accurate characterization and effective monitoring of system behavior and risk evolution [16]. However, previous studies have primarily focused on processing grey and fuzzy source information, often neglecting the ambiguity inherent in human subjective judgment when assigning weights to various indices. This oversight can result in significant discrepancies between risk assessment outcomes and actual conditions. To address this issue, a method was developed to determine indicator weights, thereby resolving the membership relationship between individual risk indicators and the overall risk status of a region. Based on

the aforementioned problems and research methods, while fully accounting for the inherent risks in urban areas, this paper integrates information diffusion theory with the grey clustering evaluation method to propose a novel approach for assessing the coupling risk of accidents in urban environments. In contrast to traditional risk assessment methods, this approach effectively addresses the issue of inaccurate parameter estimation when sample sizes are limited. Furthermore, by leveraging the distinct advantages of the grey clustering algorithm in handling "poor information", the collaborative analysis of qualitative and quantitative data is achieved through whitening functions, thereby enhancing the precision and reliability of risk assessment outcomes. The development of a regional coupling risk assessment method is anticipated to offer guidance for the rational planning of industrial parks within urban areas and the dynamic management of regional risks.

## 1 The theoretical foundation for the dynamic management and control of urban regional coupling risk

The analysis of urban area security reveals that the urban security system is a complex, expansive, and open system, composed of numerous subsystems, each with specific functions. A clear nonlinear relationship exists between the influencing factors within these subsystems. In the event of an incident in one subsystem, the significant energy released may propagate to adjacent systems, potentially triggering a catastrophic chain reaction. If the released energy exceeds the safety thresholds of the urban area's protective measures, it can compromise the integrity of the protective system, thereby amplifying the societal impact of the accident.

Therefore, this paper proposes a theoretical framework for assessing coupling risks in urban areas, utilizing a "triangle" approach (Fig 1).

The model encompasses various production systems within the region, primarily reflecting the inherent risks associated with production safety accidents. Surrounding the model are elements such as emergencies, disaster-bearing carriers, and regional emergency management systems, offering a comprehensive representation of the region's security protection system's vulnerability to disaster impacts. Building on these two layers, we construct a "triangle" correlation model to depict the dynamic coupling risks in urban areas and utilize a risk matrix for the visual analysis of the coupling outcomes between inherent risk and vulnerability in these settings.

## 2 Inherent risk analysis of production safety accidents in urban areas

### 2.1 The establishment of information diffusion model

In the numerical processing of production safety accident risk in urban areas, challenges such as information asymmetry and data scarcity—commonly referred to as the small sample problem—are often encountered. To address this, fuzzy set theory views probability distribution as a mapping from events to probabilities. By applying set-valued fuzzy mathematics,

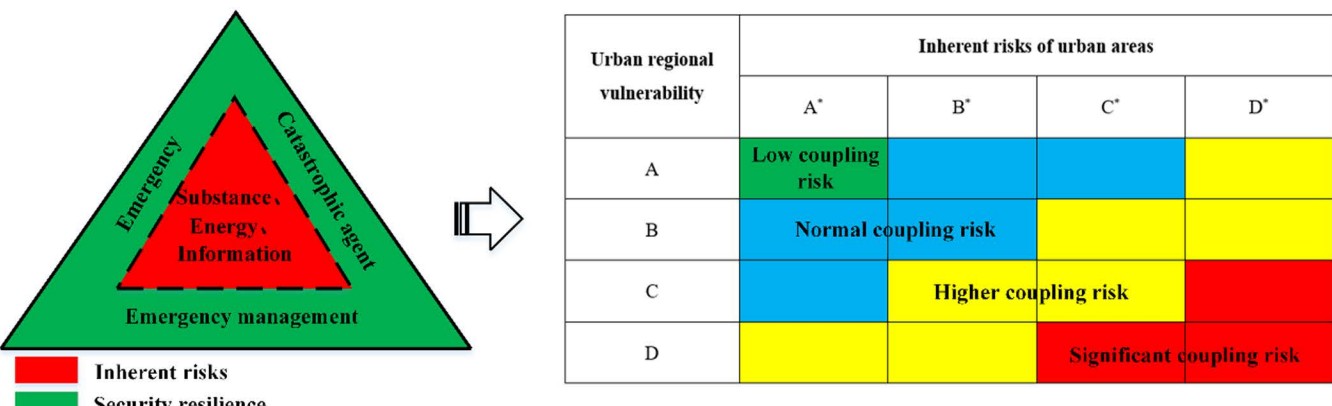

**Fig 1. "Triangle" model of coupled risks in urban areas.**

individual sample points can be processed to establish an information diffusion model. The diffusion function facilitates the establishment of binary relationships between variables, allowing for the appropriate expansion of incomplete information and compensating for the limitations posed by small sample sizes.

The fundamental principles of the information diffusion model are as follows:

**Principle 1:** In the context of a nonlinear relationship, any sample with a size smaller than the total population is considered an incomplete sample.

**Principle 2:** Given a sample $X$, let $A$ represent the true relationship of the object. The sample $X$ is considered a correct dataset for $A$, if and only if, there exists a model relationship $D$ such that when $X$ is processed through $D$, the resulting estimate of $A_X^D$ satisfies the condition $A_X^D = A$.

**Principle 3:** Given a sample $X = \{ x_k | k = 1, 2, \cdots, n \}$, let $V$ be a subset of $U$, $\mu : X \times V \to [0, 1]$, $(x, v) \to \mu(x, v)$, referred to as the information diffusion of $X$ on $V$. If the diffusion is decreasing and $\forall x \in X, \forall v', v'' \in V$, then for $\|v' - x\| \leq \|v'' - x\|$, $\mu(x, v') \geq \mu(x, v'')$, $\mu$ is defined as the information function, and $V$ is designated as the monitoring space.

**Principle 4:** The set $\mu(x, u)$ satisfies the conservation condition, if and only if $\forall x \in X$, meaning its integral over the domain $U$ equals 1. If the domain of the random variable is discrete, let $U = \{ \mu_1, \mu_2, \mu_3, \cdots, \mu_m \}$, and the conservation condition is expressed as follows.

$$\sum_{j=1}^{m} \mu(x, \mu_j) = 1 \quad \forall x \in X$$

(1)

**Principle 5:** Given a sample $X$, let $\mu(x, u)$ represent the information diffusion of $X$ over its universal set $U$. For a specific $x$, a fuzzy set is defined on $U$ by the membership function $\mu(x, u)$. For all $x$, the fuzzy set $D(x) = \{ \mu(x, u) | x \in X, u \in U \}$, composed of $\mu(x, u)$, is referred to as the fuzzy set sample of $X$ on $U$, formed by $\mu(x, u)$.

## 2.2 Extension analytic hierarchy process

This paper fully considers the inherent ambiguity in human judgment when evaluating indicators and introduces a novel index weight calculation method by integrating the theory of extenics with the Analytic Hierarchy Process (AHP). This new method, termed the Extension Analytic Hierarchy Process (EAHP), addresses both the order-preserving and consistency requirements of the scale. The 10/10–18/2 fractional scale method is employed, with the specific scale values detailed in Table 1.

The definition of EAHP is as follows.

**Definition 1:** $A = (a_{ij})_{n \times n}$ is an extension interval number judgment matrix, if $\forall i, j = 1, 2, \cdots, n$, where 1) $a_{ij} = \left\langle a_{ij}^-, a_{ij}^+ \right\rangle$ and $\frac{2}{18} \leq a_{ij}^- \leq a_{ij}^+ \leq \frac{18}{2}$; 2) $a_{ij} = \frac{1}{a_{ji}}$; 3) $a_{ji} = a_{ij}^{-1} = \left\langle \frac{1}{a_{ij}^+}, \frac{1}{a_{ij}^-} \right\rangle$; 4) $a = \langle a^-, a^+ \rangle, b = \langle b^-, b^+ \rangle$ are two extension interval numbers, then the possibility degree value of $a \geq b$ is defined as: $V(a \geq b) = \frac{2(a^+ - b^-)}{(b^+ - b^-) + (a^+ - a^-)}$.

**Definition 2:** For the f-th layer comprehensive extension interval number judgment matrix $A = (A^-, A^+)$, the steps for determining the weight vector that satisfies the consistency condition are as follows.

1) The normalized eigenvectors with positive components corresponding to the maximum eigenvalues of $A^-$ and $A^+$ are $x^-$ and $x^+$ respectively.

2) Solve the weight vector.

$$s^f = \left( s_1^f, s_2^f, \cdots, s_{n_f}^f \right)^T = \langle k x^-, m x^+ \rangle$$

(2)

In the formula (2), $k = \sqrt{\sum_{j=1}^{n_f} \frac{1}{\sum_{i=1}^{n_f} a_{ij}^+}}$, $m = \sqrt{\sum_{j=1}^{n_f} \frac{1}{\sum_{i=1}^{n_f} a_{ij}^-}}$

**Definition 3:** Hierarchy single sort, calculate $V\left(s_i^f \geq s_j^f\right)$ $(i = 1, 2, \cdots, n_f; i \neq j)$, if $\forall i, j = 1, 2, \cdots, n_f (i \neq j)$, $V\left(s_i^f \geq s_j^f\right) \geq 0$, then:

$$P_{jh}^f = 1, P_{ih}^f = V\left(s_i^f \geq s_j^f\right) \ i = 1, 2, \cdots, n_f; \ i \neq j \tag{3}$$

Among them, $P_{ih}^f$ represents the single ranking of the i-th factor on the f-th layer to the h-th factor on the f-1-th layer. After normalization, $P_h^f = \left(P_{1h}^f, P_{2h}^f, \cdots, P_{n_f h}^f\right)^T$ represents the hierarchical single ranking weight vector of each factor on the f-th layer to the h-th factor on the f-1-th layer.

**Definition 4:** Hierarchical total ranking, $n_k$ represents the number of factors in each layer. If the ranking weight vector $W^{f-1} = \left(W_1^{f-1}, W_2^{f-1}, \cdots, W_{n_k}^{f-1}\right)^T$ of the f-1 layer to the total target is , then the composite ranking $W^f$ of all elements on the f-th layer to the total target is given by the following formula.

$$W^f = \left(W_1^f, W_2^f, \cdots, W_{n_k}^f\right)^T = P^f W^{f-1} \tag{4}$$

## 2.3 The inherent risk calculation model of urban area based on information diffusion

In a closed system, the information diffusion function $\mu(x, u)$ is decreasing, meaning that information propagates in the direction of decreasing concentration. Similar to the natural diffusion of molecules, information diffusion functions to fill gaps and move toward lower concentration areas. Consequently, it can be assumed that the process of information diffusion follows Fick's First and Second Laws of molecular diffusion. Let $X = \{x_1, x_2, \cdots, x_n\}$ represent the observed samples of accident risk in a specific urban area, with the sample domain $U = \{u_1, u_2, \cdots, u_m\}$. For each observed sample point in $X$, the information it contains can be diffused to all monitoring points within its domain $U$. For the i-th sample point $x_i$, based on the normal diffusion function governing the information diffusion process, it can be inferred that:

$$f_i(u_j) = \frac{1}{h\sqrt{2\pi}} \exp\left[-\frac{(x_i - u_j)^2}{2h^2}\right] \tag{5}$$

In formula, $h$ is the diffusion coefficient, The diffusion coefficient $h$ has a specific functional relationship with the number of observed samples [17]. The specific calculation is as follows. In the formula, $n$, $a$ and $b$ are the minimum and maximum values of sample size and sample observation value, respectively.

**Table 1. 10/10–18/2 scaling.**

| Scale | Meaning (element $a_i$ compared with $a_j$) |
|---|---|
| 10/10 | The former is as important as the latter. |
| 12/8 | The former is slightly more important than the latter. |
| 14/6 | The former is more important than the latter. |
| 16/4 | The former is more important than the latter obviously |
| 18/2 | The former is most important |
| 11/9、13/7、15/5、17/3 | The median value of the above two adjacent judgments |
| The reciprocal of the above value | If factor $i$ compared with $j$ is $a_{ij}$, then factor $i$ compared with $j$ is $a_{ji} = 1/a_{ij}$ |

$$h = \begin{cases} 0.8146(b-a)/(n-1) & n = 5 \\ 0.5690(b-a)/(n-1) & n = 6 \\ 0.4560(b-a)/(n-1) & n = 7 \\ 0.3860(b-a)/(n-1) & n = 8 \\ 0.3362(b-a)/(n-1) & n = 9 \\ 0.2986(b-a)/(n-1) & n = 10 \\ 2.6851(b-a)/(n-1) & n \geq 11 \end{cases} \tag{6}$$

The $f_i(u_j)$ is normalized, $\mu_{x_i}(\mu_j) = f_i(\mu_j) / \sum\limits_{k=1}^{m} f_i(\mu_k)$. At this time, the single-valued sample point $x_i$ becomes a fuzzy subset with $\mu_{x_i}(\mu_j)$ as the membership function through information diffusion.

Let $q(\mu_j) = \sum\limits_{i=1}^{n} \mu_{x_i}(\mu_j)$, then the probability of the sample point $x_i$ falling at $\mu_j$ is estimated to be $P(\mu_j)$.

$$P(\mu_j) = q(\mu_j) / \sum\limits_{j=1}^{m} q(\mu_j) \tag{7}$$

At the moment, the probability estimate exceeds $\mu_j$, that is, the exceedance probability is $P'(\mu_j)$.

$$P'(\mu_j) = \sum\limits_{k=j}^{m} P(\mu_k) \tag{8}$$

## 3 Vulnerability analysis of urban area

### 3.1 Urban regional vulnerability

Vulnerability refers to the characteristics of objects that make them susceptible to harm, damage, or disruption. It considers the degree of risk exposure, susceptibility, and resilience. When applied to the context of urban regional safety protection systems, vulnerability is defined as the capacity of these systems to maintain their core safety characteristics and functions when subjected to external accidents and disasters. Consequently, a city should be able to adapt to uncertainty, endure stress, and recover swiftly in the face of adverse events. Specifically, when the impact of an accident is minor, the city should be able to absorb the shock; when faced with moderate accident intensity, the city should weaken the impact. In cases of severe accident intensity, the city should withstand the shock and recover rapidly through external means. This capacity reflects the vulnerability of urban regional security, as illustrated in Fig 2.

### 3.2 Urban regional vulnerability assessment

When an evaluation object is influenced by multiple index factors, accurately assessing the impact of each indicator can be challenging. To address this complexity, the author applies grey clustering theory to establish a unified grey class set for the evaluation indices and generates the corresponding grey class whitening functions. By evaluating the whitening numbers of the attributes associated with the evaluation object, the grey class of the object can be determined. Additionally, in scenarios where the evaluation object is influenced by numerous index factors, the grey clustering transformation weight calculation method is commonly used to assign attribute index weights. However, this approach can obscure the information of attributes with particularly small grey thresholds, potentially distorting the evaluation results. To mitigate this issue, this paper introduces methods and theories from matter-element extension to resolve these contradictions and proposes the matter-element extension analytic hierarchy process (EAHP) model, which effectively calculates the attribute index weights of the evaluation object. The matter-element extension grey clustering evaluation method developed in this study plays a crucial role in determining the vulnerability level of the evaluation object. The detailed evaluation process is illustrated in Fig 3.

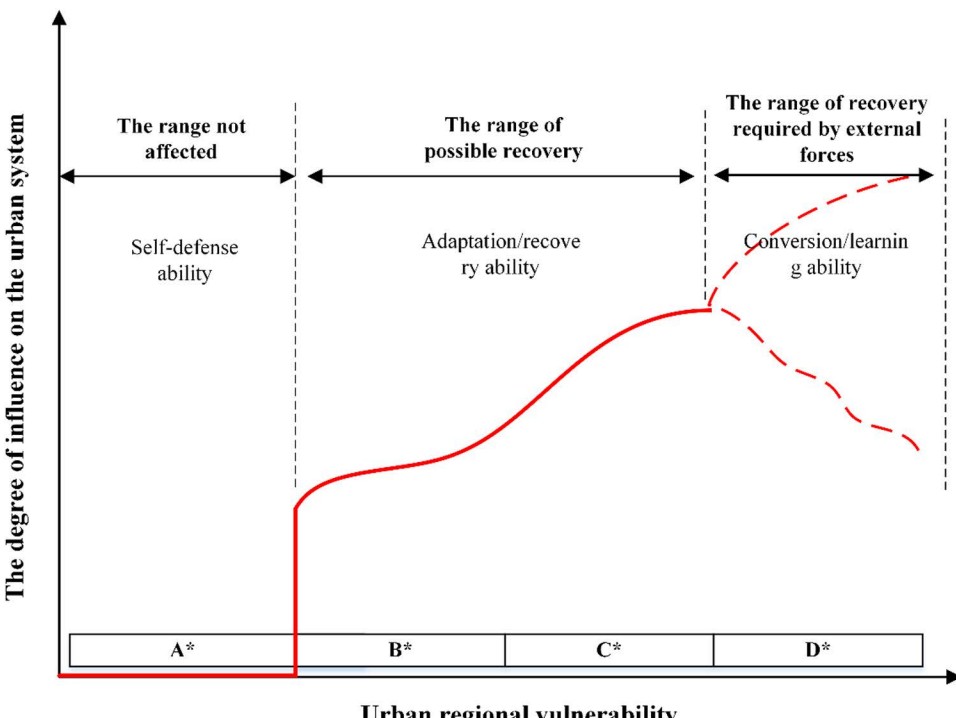

**Fig 2. Different development stages of urban vulnerability.**

**3.2.1 Construction of urban regional vulnerability assessment index system.** The higher the vulnerability of urban areas, the weaker their ability to withstand the impact of external accidents and disasters. Guided by the principles of scientific rigor, relevance, systematic approach, and data availability, and considering the characteristics of urban development in China, the urban vulnerability assessment index system is constructed from four key aspects: urban population density, professional rescue site density, emergency material reserve point density and residents' emergency disposal ability, the vulnerability classification criteria for each index are established, as presented in Table 2, in conjunction with the expertise of professionals in related industries.

**3.2.2 Vulnerability assessment model of urban area.** According to the extension theory of urban vulnerability assessment, the first step is to analyze the constituent elements and determine the matter-element expression of the object under evaluation. This matter-element expression is defined as an ordered triplet consisting of the object $A$, the characteristic $C$, and the characteristic value $V$.

$$M_A = (A_i, C_k, V_{ki}) = \begin{pmatrix} A_i & C_1 & V_{1i} \\ & C_2 & V_{2i} \\ & \vdots & \vdots \\ & C_n & V_{ni} \end{pmatrix}$$

(1) Dimensionless processing of characteristic index

Before calculating the grey clustering correlation function value, it is essential to perform dimensionless processing on the original values due to significant variations in magnitude, dimension, and orientation of the advantages and disadvantages associated with each index.

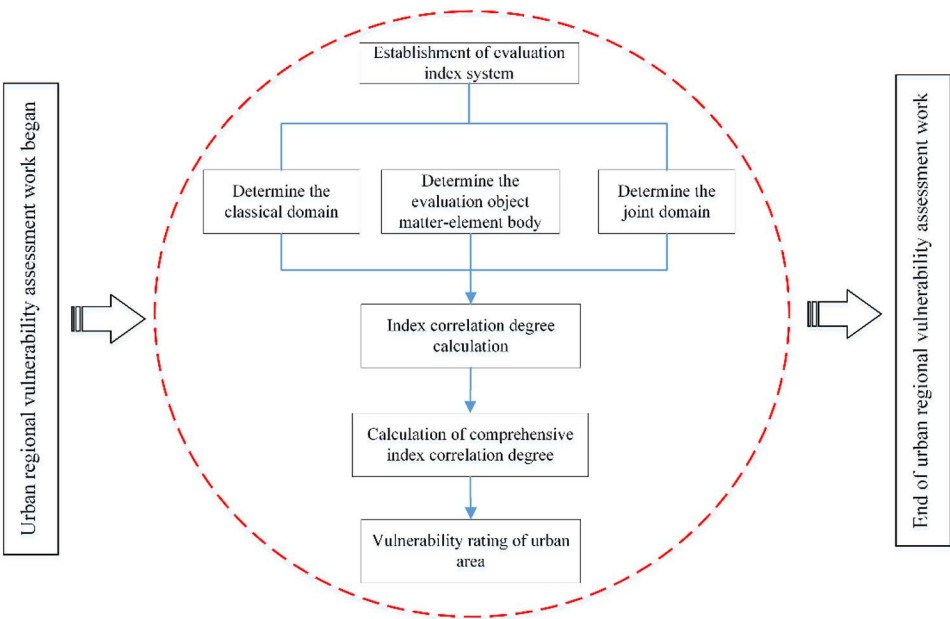

**Fig 3. Vulnerability assessment model for urban areas.**

**Table 2. Analysis index of vulnerability assessment of urban areas and judgment standard.**

| Vulnerability degree | Vulnerability | Analysis index | | | |
|---|---|---|---|---|---|
| | | Urban population density $C_1$/ (person·km$^{-2}$) | Density of professional rescue sites $C_2$/(units·10$^{-1}$ km$^{-2}$) | Density of emergency supplies reserve points $C_3$/(units·10$^{-2}$ km$^{-2}$) | Residents' capacity for emergency response $C_4$/ % (Percentage of professional trainees) |
| A* | Very strong ability to reduce risk | [0, 100] | (7, 10] | (20, 30] | (50, 100] |
| B* | Strong ability to reduce risk | (100, 2000] | (4, 7] | (15, 20] | (20, 50] |
| C* | General risk resilience capacity | (2000, 5000] | (2, 4] | (5, 15] | (10, 20] |
| D* | Poor ability to resist risk | (5000, 10000] | [0, 2] | [0, 5] | [0, 10] |

For the benefit indicators:

$$v'_{ji} = \frac{v_{ji} - \min_{j}(v_{ji})}{\max_{j}(v_{ji}) - \min_{j}(v_{ji})}$$

(9)

For the cost-based indicators:

$$v'_{ki} = \frac{\max_{k}(v_{ki}) - v_{ki}}{\max_{k}(v_{ki}) - \min_{k}(v_{ki})}$$

(10)

(2) Determination of grey whitening function $f_{nm}(c_n)$

The attribute set of the evaluation object $A$ is denoted by $C = \{c_1, c_2, \cdots, c_n\}$, where $c_n$ represents a positive value and serves as the evaluation index set for object $A$. Let $U = \{u_1, u_2, \cdots, u_m\}$ represent the linguistic grey class set, and $\lambda_{nm}$ denote the threshold value of the whitening function corresponding to the $n$-th index within the $u_m$-th grey class. To facilitate practical application and simplify the calculation of each index's whitening function, therefore, we set $m = 4$, based on the risk resistance ability, we define $U(risk\ resisting\ ability) = \{u_1 = '' strong'', u_2 = '' relatively\ strong'', u_3 = '' general'', u_4 = '' poor''\}$ as the semantic grey class set, with $f_{nm}(c_n)$ being a linear function representing the grey class whitening function. For the $n$-th index, if the threshold values are given as $\lambda_{n1}, \lambda_{n2}, \lambda_{n3}$ and $\lambda_{n4}$, such that $\lambda_{n1} > \lambda_{n2} > \lambda_{n3} > \lambda_{n4}$, the whitening function value of object $A$ can be calculated according to Eq.(11)-Eq.(14).

$$f_{n1}(c_n) = \begin{cases} 1 & v_{ni} > \lambda_{n1} \\ \frac{v_{ni} - \lambda_{n2}}{\lambda_{n1} - \lambda_{n2}} & \lambda_{n2} < v_{ni} \leq \lambda_{n1} \\ 0 & v_{ni} < \lambda_{n2} \end{cases}$$

(11)

$$f_{n2}(c_n) = \begin{cases} 0 & v_{ni} \geq \lambda_{n1} \\ \frac{\lambda_{n1} - v_{ni}}{\lambda_{n1} - \lambda_{n2}} & \lambda_{n2} \leq v_{ni} < \lambda_{n1} \\ \frac{v_{ni} - \lambda_{n3}}{\lambda_{n2} - \lambda_{n3}} & \lambda_{n3} \leq v_{ni} < \lambda_{n2} \\ 0 & v_{ni} < \lambda_{n3} \end{cases}$$

(12)

$$f_{n3}(c_n) = \begin{cases} 0 & v_{ni} \geq \lambda_{n2} \\ \frac{\lambda_{n2} - v_{ni}}{\lambda_{n2} - \lambda_{n3}} & \lambda_{n3} \leq v_{ni} < \lambda_{n2} \\ \frac{v_{ni} - \lambda_{n4}}{\lambda_{n3} - \lambda_{n4}} & \lambda_{n4} \leq v_{ni} < \lambda_{n3} \\ 0 & v_{ni} < \lambda_{n4} \end{cases}$$

(13)

$$f_{n4}(c_n) = \begin{cases} 0 & v_{ni} > \lambda_{n3} \\ \frac{v_{ni} - \lambda_{n3}}{\lambda_{n3} - \lambda_{n4}} & \lambda_{n4} < v_{ni} \leq \lambda_{n3} \\ 1 & v_{ni} < \lambda_{n4} \end{cases}$$

(14)

The vulnerability of urban areas is assigned by indicators. Following the conventional percentage assignment method, higher values correspond to greater urban resilience and lower vulnerability. Table 3 presents the dimensionless processing, specific threshold settings, and their corresponding relationship with the grey class based on the relevant content in Table 2.

(3) Grey clustering analysis of evaluation object

The clustering membership degree of evaluation object $A$ with respect to the $u_m$ gray class is calculated using the formula (15).

$$\gamma_m = \sum_{n-1}^{4} f_{nm}(c_n) \times w_n \qquad m = 1, 2, 3, 4$$

(15)

In the formula, $w_n$ is the weight of n-th attribute characteristic indexes of the evaluation object, and its value is calculated by the EAHP method.

**Table 3. Grey whitening function threshold of urban regional vulnerability evaluation index.**

| Vulner-ability degree | Vulnerability | Analysis index | | | | | |
|---|---|---|---|---|---|---|---|
| | | Urban population densit y $C_1$/(person·km$^{-2}$) | | | Density of professional rescue sites $C_2$/(units·10$^{-1}$ km$^{-2}$) | | |
| | | Dimensionless processing of judgment standard | Thresh-old setting | Index assignment | Dimensionless processing of judgment standard | Thresh-old setting | Index assignment |
| A* | Very strong ability to reduce risk | [0.99,1] | 90 | $\frac{v_{ki}-0.99}{1-0.99} \times (100-90)+90$ | (0.7,1] | 90 | $\frac{v_{ki}-0.7}{1-0.7} \times (100-90)+90$ |
| B* | Strong ability to reduce risk | (0.8,0.99] | 80 | $\frac{v_{ki}-0.8}{0.99-0.8} \times (90-80)+80$ | (0.4,0.7] | 80 | $\frac{v_{ki}-0.4}{0.7-0.4} \times (90-80)+80$ |
| C* | General risk resilience capacity | (0.5,0.8] | 70 | $\frac{v_{ki}-0.5}{0.8-0.5} \times (80-70)+70$ | (0.2,0.4] | 70 | $\frac{v_{ki}-0.2}{0.4-0.2} \times (80-70)+70$ |
| D* | Poor ability to resist risk | [0,0.5) | 60 | $\frac{v_{ki}}{0.5} \times 60+60$ | [0,0.2] | 60 | $\frac{v_{ki}}{0.2} \times 60+60$ |
| Vulner-ability degree | Vulnerability | Analysis index | | | | | |
| | | Density of emergency supplies reserve points $C_3$/(units·10$^{-2}$ km$^{-2}$) | | | Residents' capacity for emergency response $C_4$/ % (Percentage of professional trainees) | | |
| | | Dimensionless processing of judgment standard | Thresh-old setting | Index assignment | Dimensionless processing of judgment standard | Thresh-old setting | Index assignment |
| A* | Very strong ability to reduce risk | (0.67,1] | 90 | $\frac{v_{ki}-0.67}{1-0.67} \times (100-90)+90$ | (0.5,1] | 90 | $\frac{v_{ki}-0.5}{1-0.5} \times (100-90)+90$ |
| B* | Strong ability to reduce risk | (0.5,0.67] | 80 | $\frac{v_{ki}-0.5}{0.67-0.5} \times (90-80)+80$ | (0.2,0.5] | 80 | $\frac{v_{ki}-0.2}{0.5-0.2} \times (90-80)+80$ |
| C* | General risk resilience capacity | (0.17,0.5] | 70 | $\frac{v_{ki}-0.17}{0.5-0.17} \times (80-70)+70$ | (0.1,0.2] | 70 | $\frac{v_{ki}-0.1}{0.2-0.1} \times (80-70)+70$ |
| D* | Poor ability to resist risk | [0,0.17] | 60 | $\frac{v_{ki}}{0.17} \times 60+60$ | [0,0.1] | 60 | $\frac{v_{ki}}{0.1} \times 60+60$ |

According to Eqs.(11)-(14), the whitening function diagrams corresponding to the vulnerability levels A

*, B

*, C

*, and D

*can be drawn, as shown in Fig 4.

Establish the gray clustering membership sequence of the evaluation object $A$, $\gamma_m = \{\gamma_1, \gamma_2, \gamma_3, \gamma_4\}$, define $\gamma_m^* = \max\{\gamma_1, \gamma_2, \gamma_3, \gamma_4\}$, so as to determine that the evaluation object belongs to the gray class $\gamma_m^*$.

## 4 Case analysis

District A of the city serves as a pivotal economic development center, particularly renowned for its concentration of indus-trial production and high-tech industries. Currently encompassing 14 streets and 6 townships, District A has experienced rapid economic growth in recent years. However, the prevailing industrial landscape dominated by chemical industry, mechanical manufacturing, electronic information processing, etc., exposes the region to heightened risks with occasional safety production accidents. This study employs four commonly used indicators in China's safety production statistics

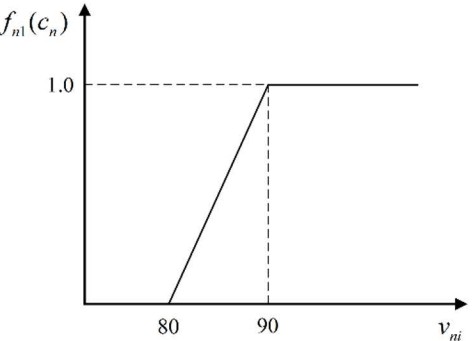

(1) Whitening function diagram of vulnerability index at level A°

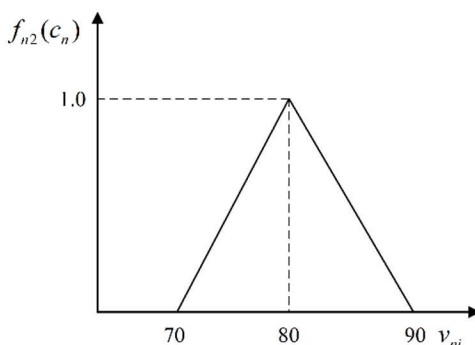

(2) Whitening function diagram of vulnerability index at level B°

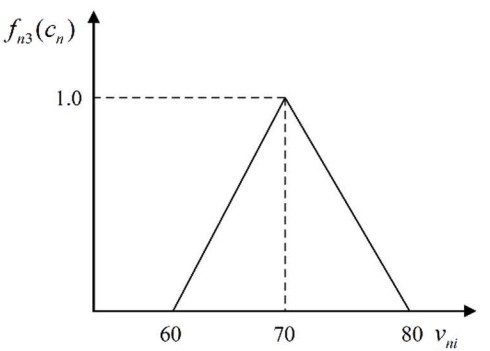

(3) Whitening function diagram of vulnerability index at level C°

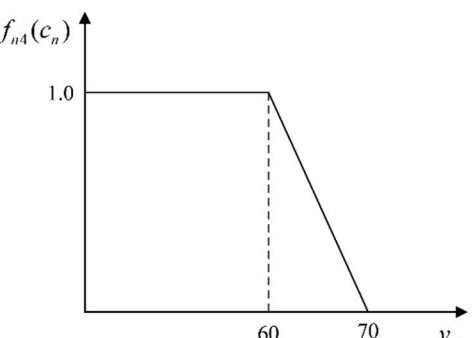

(4) Whitening function diagram of vulnerability index at level D°

**Fig 4. Whitenization function diagram of vulnerability level.**

**Table 4. Sample raw indicators.**

| Year | Evaluation index | | | |
|---|---|---|---|---|
| | Construction death/person | Fire death/ person | The death of industrial and mining commerce/person | Billion yuan GDP mortality rate/ $\times 10^{-2}$ |
| 2018 | 8 | 4 | 4 | 10.26 |
| 2019 | 4 | 3 | 4 | 8.98 |
| 2020 | 3 | 1 | 1 | 7.08 |
| 2021 | 5 | 2 | 1 | 4.89 |
| 2022 | 3 | 1 | 2 | 2.94 |
| 2023 | 2 | 0 | 3 | 1.59 |

- construction mortality rate, fire mortality rate, industrial and mining trade mortality rate, and billion yuan GDP mortality rate - to evaluate inherent risks within urban areas. The statistical data for each index in Area A from 2018 to 2023 are selected as the original data sample (Table 4).

## 4.1 Inherent risk calculation of urban areas

**1) Sample information diffusion.** The four sample spaces representing the annual construction mortality rate, fire mortality rate, industrial and mining trade mortality rate, and mortality rate per billion yuan of GDP in this area are denoted

as $X_1$, $X_2$, $X_3$, and $X_4$, respectively. The one-dimensional space sets [0,9], [0,6], [0,6], and [0,12] are considered as the domains for $X_1$, $X_2$, $X_3$, and $X_4$, respectively. The upper limit of each set is determined by the actual maximum statistical value observed for each sample. Furthermore, the step size ($\Delta$) for discretization is set to be 3 for $X_1$, and 2 for both $X_2$ and $X_3$, while it is 4 for $X_4$. The inherent risk levels associated with these samples range from low to high, corresponding to categories $A$, $B$, $C$, and $D$, respectively. Consequently, the discrete domains can be defined as follows.

$U_1 = \{0, 3, 6, 9\}$, total of four risk levels; $U_2 = \{0, 2, 4, 6\}$, total of four risk levels; $U_3 = \{0, 2, 4, 6\}$, total of four risk levels; $U_4 = \{0, 4, 8, 12\}$, total of four risk levels.

According to formula (6), the diffusion coefficients of the four samples can be calculated as follows: $h_1 = 1.0242$, $h_2 = 0.6828$, $h_3 = 0.6828$, and $h_4 = 1.3656$. By utilizing Matlab for auxiliary calculations based on formulas (5) to (8), we can obtain the corresponding fuzzy membership function for each sample, as well as the probabilities and exceedance probabilities associated with different risk levels for each sample. The specific calculation results are presented in Table 5.

According to the analysis of the calculation results in Table 5, the risk estimates of each index decrease with the increase of the risk level. The maximum probability of the annual mortality rate of construction to constitute the B-level risk level is 0.6254, the maximum probability of the annual mortality rate of fire to constitute the B-level risk level is 0.4167, the maximum probability of the annual mortality rate of industry and mining to constitute the B-level risk level is 0.4166, and the maximum probability of the mortality rate of billion yuan GDP to constitute the C-level risk level is 0.3760.

**2) Risk assessment index weight calculation.** The inherent risk assessment index $X_1 \sim X_4$ of safety production in urban areas were used as the comparison indices for rows and columns, respectively. The comparative analysis results were then filled into table according to the developed score scale method (Table 1). Consequently, a score scale determination table for the inherent risk assessment index of safety production in urban areas was constructed, as presented in Table 6.

According to the index score scale judgment table, the extension interval number judgment matrix can be obtained, $A = (A^-, A^+)$.

**Table 5. Estimated risk values for each indicator in Zone A.**

| Index $Xi$ | Risk level $uj$ | Probability $P(\mu j)$ | Exceedance probability $P'(\mu j)$ |
|---|---|---|---|
| Annual construction mortality rate $X_1$ | A | 0.0367 | 1 |
| | B | **0.6254** | 0.9633 |
| | C | 0.2033 | 0.3379 |
| | D | 0.1346 | 0.1346 |
| Annual fire mortality rate $X_2$ | A | 0.3333 | 1 |
| | B | **0.4167** | 0.6667 |
| | C | 0.2478 | 0.2500 |
| | D | 0.0022 | 0.0022 |
| Annual mortality rate of industrial and mining commerce $X_3$ | A | 0.1689 | 1 |
| | B | **0.4166** | 0.8311 |
| | C | 0.4100 | 0.4145 |
| | D | 0.0045 | 0.0045 |
| Billion yuan GDP mortality rate $X_4$ | A | 0.1377 | 1 |
| | B | 0.3632 | 0.8623 |
| | C | **0.3760** | 0.4991 |
| | D | 0.1231 | 0.1231 |

**Table 6. Evaluation indicator score scale determination table.**

| Xi | X₁ | X₂ | X₃ | X₄ |
|---|---|---|---|---|
| $X_1$ | <1, 1> | <6/14, 7/13> | <4/16, 5/15> | <2/18, 3/17> |
| $X_2$ | <13/7, 14/6> | <1, 1> | <7/13, 8/12> | <6/14, 7/13> |
| $X_3$ | <15/5, 16/4> | <12/8, 13/7> | <1, 1> | <8/12, 9/11> |
| $X_4$ | <17/3, 18/2> | <13/7, 14/6> | <11/9, 12/8> | <1, 1> |

$$A^- = \begin{pmatrix} 1 & 6/14 & 4/16 & 2/18 \\ 13/7 & 1 & 7/13 & 6/14 \\ 15/5 & 12/8 & 1 & 8/12 \\ 17/3 & 13/7 & 11/9 & 1 \end{pmatrix}$$

$$A^+ = \begin{pmatrix} 1 & 7/13 & 5/15 & 3/17 \\ 14/6 & 1 & 8/12 & 7/13 \\ 16/4 & 13/7 & 1 & 9/11 \\ 18/2 & 14/6 & 12/8 & 1 \end{pmatrix}$$

According to Eq. (2), it can be calculated that, $k = 0.9572$, $m = 1.0398$.

Finally, the weight vector of the inherent risk index of safety production in urban areas is obtained by applying the formulas (3) and (4), followed by normalization, $W = (0.0207, 0.2889, 0.4184, 0.272)$.

The weight values of each risk assessment index derived from the analysis are calculated using the consistency ratio (CR), yielding CR = 0.0318 < 0.1. This result indicates that the judgment matrix satisfies the consistency requirements.

**3) Reassessment of the inherent risk in Area A.** The inherent risk results of District A can be obtained through calculation and analysis, based on the risk estimation value of each index and the weight assigned to evaluation indices, as presented in Table 7.

The analysis of the calculation results reveals that when A district in a city is at level $B$, the probability of inherent risk is 0.406426. Therefore, it can be inferred that the overall inherent risk in this area remains at a $B$-level.

### 4.2 Vulnerability analysis of urban area

**1) Matter element representation of evaluation object.**

$$M_A = (A_i, C_k, V_{ki}) = \begin{pmatrix} A_i & C_1 & V_{1i} \\ & C_2 & V_{2i} \\ & \vdots & \vdots \\ & C_n & V_{ni} \end{pmatrix} = \begin{pmatrix} AreaA & C_1 & 1628 \\ & C_2 & 6 \\ & C_3 & 14 \\ & C_4 & 30 \end{pmatrix}$$

**Table 7. Inherent risk calculation results of Zone A.**

| Risk level $U_i$ | Probability $P(\mu_i)$ | Exceedance probability $P'(\mu_i)$ |
|---|---|---|
| A | 0.205172 | 1 |
| B | **0.406426** | 0.6116 |
| C | 0.349614 | 0.9612 |
| D | 0.038788 | 0.038788 |

**Table 8. Vulnerability assessment indicator statistics of Zone A.**

| Characteristic indexes | Urban population density $C_1$/ (person·km$^{-2}$) | Density of professional rescue sites $C_2$/(units·10$^{-1}$ km$^{-2}$) | Density of emergency supplies reserve points $C_3$/(units·10$^{-2}$ km$^{-2}$) | Residents' capacity for emergency response $C_4$/ % (Percentage of professional trainees) |
|---|---|---|---|---|
| Statistical data | 1628 | 6 | 14 | 30 |
| Dimensionless analysis | 0.84 | 0.6 | 0.47 | 0.30 |
| Index assignment | 82.10 | 86.67 | 79.09 | 83.33 |

**2) Dimensionless manipulation of characteristic indices.** According to the statistical data obtained from field investigation (Table 8), a dimensionless analysis of the characteristic index value is conducted using Formulas (9) and (10).

**3) Vulnerability assessment of Area A.** The vulnerability assessment of Area A is conducted based on Formulas (11) - (14), and the clustering correlation between each vulnerability assessment index and its corresponding vulnerability level is determined, as shown in Table 9.

By employing the similar methodology for computing the weight of the inherent risk index, we derived the weight vector $D = (0.0207, 0.2889, 0.4184, 0.272)$ for evaluating the vulnerability analysis index in Area **A**.

According to Formula (15), the vulnerability analysis results of Area A can be derived by integrating the calculation outcomes of the correlation degree between vulnerability analysis indices and their corresponding weights, as illustrated in Table 10.

The analysis of the calculation results reveals that the region exhibits a *B*-level vulnerability, indicating a robust capacity to withstand risks.

**4) Urban regional coupling risk assessment.** The area's inherent risk and vulnerability are both assessed as grade *B*. According to the coupling risk judgment matrix in Fig 1, the dynamic coupling risk of the area is classified as 'blue code', indicating a general level of risk.

### 4.3 The verification and analysis of the current safety status in Area A

Given the recent rapid economic development in District A of a city, the prevailing extensive growth model can no longer adequately meet the region's further developmental requirements. To adapt to the new normal of economic progress, it is imperative to continuously reinforce innovation-driven initiatives and gradually phase out enterprises characterized by outdated production capacity, severe pollution, and high safety risks. This transformation and upgrading of security

**Table 9. Vulnerability analysis indicator correlation calculation of Zone A.**

| Gay class value $\gamma_m$ / Evaluation oject | $\gamma_1$ | $\gamma_2$ | $\gamma_3$ | $\gamma_4$ |
|---|---|---|---|---|
| Urban population density $\omega_1 = 0.051$ | 0.21 | 0.79 | 0 | 0 |
| Density of professional rescue sites $\omega_2 = 0.117$ | 0.33 | 0.67 | 0 | 0 |
| Density of emergency supplies reserve points $\omega_3 = 0.256$ | 0 | 0.909 | 0.091 | 0 |
| Residents' capacity for emergency response $\omega_4 = 0.576$ | 0.33 | 0.67 | 0 | 0 |

**Table 10. Vulnerability risk analysis result of Zone A.**

| Vulnerability degree | A | B | C | D |
|---|---|---|---|---|
| Area A vulnerability | 0.24 | **0.74** | 0.02 | 0 |

measures have effectively enhanced both the quality and efficiency of economic advancement. In 2023, overall safety conditions within the district were commendable with a notable reduction of 17% in accidents and 33.8% in fatalities. The general risk situation remained consistent with regional dynamic risk assessment outcomes while remaining within manageable limits. In addition, the inherent risk assessment of Area A indicates a higher mortality rate per 100 million yuan of GDP. Given the presence of 289 key regulated enterprises in the region, including 132 enterprises involving hazardous chemicals, it is imperative to prioritize enhancing daily risk management in the future. Specific measures should include establishing and refining the safety production responsibility system, implementing real-time monitoring of central hazard installations, further optimizing the emergency rescue plan system, and ensuring that working environments and facilities strictly comply with relevant laws, regulations, and standards. In conjunction with routine operations, efforts should be made to strengthen the development of regional professional emergency rescue capabilities and construct an integrated emergency system that combines science and technology, management, and culture through multi-level, multi-channel, and multi-form publicity and education.

## 5 Conclusion

Amidst the rapid pace of urbanization, cities are increasingly confronted with uncertainties and unforeseen risks. The vulnerabilities of urban systems become pronounced when faced with sudden natural or anthropogenic disasters. While numerous scholars have examined urban security from specialized perspectives and in the context of individual disaster types, a comprehensive, standardized, and scientifically grounded approach to response strategies is often lacking. Establishing a safe and resilient city, underpinned by a public security framework, enables effective resistance to both internal and external risks affecting urban operations. Such cities can preserve their core structures and functions after major disruptions and facilitate rapid recovery and adaptive measures that support sustainable development. Building on this foundation, a smart, secure, and resilient city is characterized by its advanced capabilities for scientific identification, comprehensive situational awareness, and intelligent response. To harmonize urban development with security imperatives, it is crucial to enhance the construction of resilient cities through scientific and technological innovation and improved management practices. Simultaneously, fostering a culture that addresses critical barriers to modernizing urban public security governance is essential to ensuring public safety.

1) The security system of urban area is a complex open giant system, which is composed of many subsystems with specific functions, and there is a nonlinear relationship between the influencing factors. Therefore, this paper proposes a "triangle model" of coupling risk in urban areas, which provides a basis for reasonable control of accident risk in urban areas.

2) In view of the situation of information asymmetry and lack of data in the process of numerical processing of production safety accident risk in urban areas, the single sample point is processed by set-valued fuzzy mathematics, and the information diffusion model is established to expand the incomplete information appropriately, so as to make up for the deficiency of small sample information.

3) The research findings have been successfully implemented in practical applications. The analysis reveals that a district in the city is classified as a 'blue code' for coupling risk, indicating an overall level of risk. Specifically, the calculated inherent risk probability for level-$B$ is 0.406426, and the vulnerability at level-$B$ suggests a high capacity to withstand risks. These analytical results are consistent with the actual circumstances.

4) This method still exhibits a degree of subjectivity in selecting indices and establishing grading criteria, necessitating further research to enhance its objectivity and scientific rigor.

## Author contributions

**Conceptualization:** Sheng Xue.

**Data curation:** Shuang Wang.

**Funding acquisition:** Chen Lv, Xiaolu Wang.

**Methodology:** Xiaolu Wang.

**Writing – original draft:** Chen Lv.

**Writing – review & editing:** Chen Lv.

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
