## [Decision Letter · Decision Letter 0]

28 Jan 2025

PONE-D-24-56332Research on Coupling Risk Assessment Method of Industrialized Urban Area Based on Information Diffusion and Extension Grey Clustering ModelPLOS ONE

Dear Dr. Lv,

Thank you for submitting your manuscript to PLOS ONE. After careful consideration, we feel that it has merit but does not fully meet PLOS ONE’s publication criteria as it currently stands. Therefore, we invite you to submit a revised version of the manuscript that addresses the points raised during the review process.

We look forward to receiving your revised manuscript.

Kind regards,

Guojin Qin

Academic Editor

PLOS ONE

Journal Requirements:

This research was supported by National Key R&D Program of China (Grant No.2021YFB3301100), Natural Science Foundation of Zhejiang Province (Grant No. LQ20E040005), the State Key Laboratories Program of China Grant No. JYBSYS2019102�.

5. Please upload a new copy of Figure 1, 2, 3, and 4 as the detail is not clear. Please follow the link for more information: "https://blogs.plos.org/plos/2019/06/looking-good-tips-for-creating-your-plos-figures-graphics/
https://blogs.plos.org/plos/2019/06/looking-good-tips-for-creating-your-plos-figures-graphics/"

Reviewers' comments:

Reviewer's Responses to Questions

**Comments to the Author**

1. Is the manuscript technically sound, and do the data support the conclusions?

Reviewer #1: Partly

Reviewer #2: Yes

2. Has the statistical analysis been performed appropriately and rigorously? 

Reviewer #1: Yes

Reviewer #2: Yes

3. Have the authors made all data underlying the findings in their manuscript fully available?

Reviewer #1: Yes

Reviewer #2: Yes

4. Is the manuscript presented in an intelligible fashion and written in standard English?

Reviewer #1: Yes

Reviewer #2: Yes

5. Review Comments to the Author

Reviewer #1: This paper proposes a dynamic control model for coupling risks in industrialized urban areas, integrating system engineering principles and focusing on two dimensions: the inherent risk of production safety accidents and the vulnerability of regional safety protection systems. The authors give detailed description of their method.

But the comparison to other methods are not enough.

It is better to explain why their method is better than others theoretically And also give the empirical test result compared with others .

Reviewer #2: Based on a detailed review of the manuscript “Research on Coupling Risk Assessment Method of Industrialized Urban Area Based on Information Diffusion and Extension Grey Clustering Model”, The coupled risk assessment method combining information diffusion and gray clustering proposed in this paper is an extension of the traditional risk assessment method, which is especially suitable for small sample data scenarios. In particular, it has strong theoretical innovation in the dynamic management of urban area risk. However, there are some weaknesses and improvements in the thesis. So I propose the following detailed critique and recommendations for major revisions.

1. The selection of diffusion coefficients for the information diffusion model (e.g., h1, h2, h3, h4 in the paper) lacks sufficient theoretical explanation and selection basis. The paper only lists the calculation process through mathematical formulas, but does not discuss the scope of application of diffusion coefficients and their influence on the results.

2. The threshold setting in the gray clustering method is too subjective, please add relevant analysis and validation.

3. No specific policy recommendations or emergency management measures based on this model are mentioned in the text after the risk level is derived.

4. Can the indicator system include all the main risk factors of industrialized urban areas? Does the author consider adding relevant indicators such as environmental risks and traffic safety?

5. How is the consistency of the EAHP methodology in the calculation of indicators ensured?

The article is recommended for publication with modification of the above issues.

6. PLOS authors have the option to publish the peer review history of their article (what does this mean? ). If published, this will include your full peer review and any attached files.

**Do you want your identity to be public for this peer review?** For information about this choice, including consent withdrawal, please see our Privacy Policy .

Reviewer #1: No

Reviewer #2: No

---

## [Author Response · Author response to Decision Letter 1]

19 Mar 2025

Dear Prof. Qin,

Thank you very much for your letter and the comments about our paper submitted to Plos One.

We have checked the manuscript and revised it according to the comments. We submit here the revised manuscript as well as a list of changes.

If you have any question about this paper, please don’t hesitate to let me know.

Sincerely yours,

Chen Lv

Response to Editor:

Thanks for your comments on our paper. We have revised our paper according to your comments:

Reply We have verified the formatting of the article against PLOS ONE's style requirements.

(2) Please note that PLOS ONE has specific guidelines on code sharing for submissions in which author-generated code underpins the findings in the manuscript. In these cases, we expect all author-generated code to be made available without restrictions upon publication of the work.

Reply This paper does not address this situation.

(3) We note that the grant information you provided in the ‘Funding Information’ and ‘Financial Disclosure’ sections do not match.

Reply According to the editor's suggestions, we have provided the “Funding Information” in cover letter.

Reply According to the editor's suggestions, we have removed the Acknowledgments section from the article and include our amended statements within our cover letter.

(5) Please upload a new copy of Figure 1, 2, 3, and 4 as the detail is not clear.

Reply According to the editor's suggestions, the resolutions of all figures have been enhanced.

Response to Reviewer 1

Thanks for your comments on our paper. We have revised our paper according to your comments:

This paper proposes a dynamic control model for coupling risks in industrialized urban areas, integrating system engineering principles and focusing on two dimensions: the inherent risk of production safety accidents and the vulnerability of regional safety protection systems. The authors give detailed description of their method.

But the comparison to other methods are not enough.

It is better to explain why their method is better than others theoretically And also give the empirical test result compared with others.

Reply According to the reviewer's suggestions, we revised the article's introduction section, providing a systematic analysis and concise summary of the limitations of the methods proposed by previous scholars. Furthermore, we elaborated on the distinctive advantages of the technique introduced in this study and highlighted its innovative aspects compared to alternative approaches.

In article

0 Introduction

……

However, in urban regional risk calculation, the unclear internal correlation characteristics often lead to missing risk sample information, thereby affecting the accuracy of risk assessment. Moreover, multi-disaster coupling risk analysis remains challenging due to issues such as large data volumes, complex data analysis, inadequate descriptions of disaster consequences, and difficulties in disaster scenario modeling. Consequently, most existing studies primarily focus on qualitative assessments of relative risk levels, while quantitative analyses of comprehensive risks associated with multiple disasters remain insufficient. From the existing research results, statistical analysis methods have demonstrated extensive applicability. Specifically, constructing logistic regression models and establishing disaster data platforms based on experiments and simulations have become typical risk analysis approaches. However, achieving a quantitative assessment of multi-hazard comprehensive risks based on data, models, and computational methods remains challenging. Key challenges include systematically identifying and integrating the critical components and core processes of various types of emergencies, clarifying interaction pathways among different elements within the disaster system, analyzing the structure and fusion techniques for multi-source data, developing an integrated framework for multi-type models, and optimizing the allocation and utilization of computational resources. Furthermore, the impact of uncertainties inherent in current assessment methods and data sources on risk analysis requires further investigation. In summary, numerous issues remain unresolved in urban regional risk assessment. Improving calculation accuracy under conditions of incomplete data and establishing a scientifically rigorous and systematic multi-hazard risk quantification method represent crucial directions for future research.

Information diffusion is a fuzzy mathematical processing method that handles single sample points using set-valued processing[15]. It establishes binary relationships between variables via a diffusion function, which appropriately extends incomplete information to compensate for the limitations caused by small sample sizes. The grey clustering evaluation method, grounded in grey system theory, extracts valuable insights from partial known information, enabling the accurate characterization and effective monitoring of system behavior and risk evolution[16]. However, previous studies have primarily focused on processing grey and fuzzy source information, often neglecting the ambiguity inherent in human subjective judgment when assigning weights to various indices. This oversight can result in significant discrepancies between risk assessment outcomes and actual conditions. To address this issue, a method was developed to determine indicator weights, thereby resolving the membership relationship between individual risk indicators and the overall risk status of a region. Based on the aforementioned problems and research methods, while fully accounting for the inherent risks in urban areas, this paper integrates information diffusion theory with the grey clustering evaluation method to propose a novel approach for assessing the coupling risk of accidents in urban environments. In contrast to traditional risk assessment methods, this approach effectively addresses the issue of inaccurate parameter estimation when sample sizes are limited. Furthermore, by leveraging the distinct advantages of the grey clustering algorithm in handling "poor information", the collaborative analysis of qualitative and quantitative data is achieved through whitening functions, thereby enhancing the precision and reliability of risk assessment outcomes. The development of a regional coupling risk assessment method is anticipated to offer guidance for the rational planning of industrial parks within urban areas and the dynamic management of regional risks.

Response to Reviewer 2

Thanks for your comments on our paper. We have revised our paper according to your comments:

(1) The selection of diffusion coefficients for the information diffusion model (e.g., h1, h2, h3, h4 in the paper) lacks sufficient theoretical explanation and selection basis. The paper only lists the calculation process through mathematical formulas, but does not discuss the scope of application of diffusion coefficients and their influence on the results.

Reply Thank you for your question. In selecting the diffusion coefficients, we drew upon research findings from the relevant field and supplemented this section with reference citations to ensure the precision and reliability of our evidence.

In article

In formula, is the diffusion coefficient, The diffusion coefficient has a specific functional relationship with the number of observed samples[17].……

References:

17 Hui S, Yueqing Z, Feng S, Liang Y. Study of urban regional risk based on information diffusion. China Safety Science Journal. 2011;21(05):166-70. doi: 10.16265/j.cnki.issn1003-3033.2011.05.028.

(2) The threshold setting in the gray clustering method is too subjective, please add relevant analysis and validation.

Reply Thank you for your valuable feedback. With regard to the threshold setting, it is indeed a limitation of the current model. At present, the threshold setting predominantly depends on the subjective judgment of experts, lacking systematic and objective criteria. In future research, we will focus on conducting in-depth investigations and making improvements to address this issue, thereby enhancing the accuracy and applicability of the model.

In article

3.2.1 Construction of urban regional vulnerability assessment index system

The higher the vulnerability of urban areas, the weaker their ability to withstand the impact of external accidents and disasters. Guided by the principles of scientific rigor, relevance, systematic approach, and data availability, and considering the characteristics of urban development in China, the urban vulnerability assessment index system is constructed from four key aspects: urban population density, professional rescue site density, emergency material reserve point density and residents' emergency disposal ability, the vulnerability classification criteria for each index are established, as presented in Table 2, in conjunction with the expertise of professionals in related industries.

5 Conclusion

……

4) This method still exhibits a degree of subjectivity in selecting indices and establishing grading criteria, necessitating further research to enhance its objectivity and scientific rigor.

(3) No specific policy recommendations or emergency management measures based on this model are mentioned in the text after the risk level is derived.

Reply Thank you for your valuable suggestions. Based on the risk assessment results, we have supplemented and refined the content of the recommended emergency management measures to ensure their comprehensiveness and applicability.

In article

4.3 The verification and analysis of the current safety status in Area A

Given the recent rapid economic development in District A of a city, the prevailing extensive growth model can no longer adequately meet the region's further developmental requirements. To adapt to the new normal of economic progress, it is imperative to continuously reinforce innovation-driven initiatives and gradually phase out enterprises characterized by outdated production capacity, severe pollution, and high safety risks. This transformation and upgrading of security measures have effectively enhanced both the quality and efficiency of economic advancement. In 2023, overall safety conditions within the district were commendable with a notable reduction of 17% in accidents and 33.8% in fatalities. The general risk situation remained consistent with regional dynamic risk assessment outcomes while remaining within manageable limits. In addition, the inherent risk assessment of Area A indicates a higher mortality rate per 100 million yuan of GDP. Given the presence of 289 key regulated enterprises in the region, including 132 enterprises involving hazardous chemicals, it is imperative to prioritize enhancing daily risk management in the future. Specific measures should include establishing and refining the safety production responsibility system, implementing real-time monitoring of central hazard installations, further optimizing the emergency rescue plan system, and ensuring that working environments and facilities strictly comply with relevant laws, regulations, and standards. In conjunction with routine operations, efforts should be made to strengthen the development of regional professional emergency rescue capabilities and construct an integrated emergency system that combines science and technology, management, and culture through multi-level, multi-channel, and multi-form publicity and education.

(4) Can the indicator system include all the main risk factors of industrialized urban areas? Does the author consider adding relevant indicators such as environmental risks and traffic safety?

Reply We extend our gratitude to the reviewers for their insightful and constructive feedback. In developing the coupled risk assessment method, we thoroughly accounted for the features of industrialized urban development, with particular attention to the influence of industrial activities on the overall risk profile of urban areas. Consequently, we prioritized the careful selection of risk factors. As the reviewers correctly noted, the set of risk factors considered in this study might not yet be fully comprehensive. The primary objective of this research is to construct a universally applicable and extensible coupled risk assessment framework. With regard to the reviewers' recommendation to incorporate additional relevant indicators, we plan to address this limitation in subsequent studies to further enhance the comprehensiveness and objectivity of the risk assessment outcomes. Once again, we sincerely appreciate the reviewers' valuable comment.

(5) How is the consistency of the EAHP methodology in the calculation of indicators ensured?

Reply Thank you for raising this valuable question. With regard to the consistency of the EAHP methodology, we have included in the article the detailed calculation results of the consistency ratio (CR) for the weight values of each risk indicator.

In article

4.1 Inherent risk calculation of urban areas

2�Risk assessment index weight calculation

……

The weight values of each risk assessment index derived from the analysis are calculated using the consistency ratio (CR), yielding CR=0.0318�0.1. This result indicates that the judgment matrix satisfies the consistency requirements.

Lastly, we extend our sincere gratitude for the exceptional and scholarly revision of our manuscript.

---

## [Decision Letter · Decision Letter 1]

27 Mar 2025

Research on Coupling Risk Assessment Method of Industrialized Urban Area Based on Information Diffusion and Extension Grey Clustering Model

PONE-D-24-56332R1

Dear Dr. Lv,

We're pleased to inform you that your manuscript has been judged scientifically suitable for publication and will be formally accepted for publication once it meets all outstanding technical requirements.

Kind regards,

Dr. Guojin Qin

Academic Editor

PLOS ONE

Additional Editor Comments (optional):

Reviewers' comments:

Reviewer's Responses to Questions

**Comments to the Author**

1. If the authors have adequately addressed your comments raised in a previous round of review and you feel that this manuscript is now acceptable for publication, you may indicate that here to bypass the “Comments to the Author” section, enter your conflict of interest statement in the “Confidential to Editor” section, and submit your "Accept" recommendation.

Reviewer #2: (No Response)

2. Is the manuscript technically sound, and do the data support the conclusions?

Reviewer #2: (No Response)

3. Has the statistical analysis been performed appropriately and rigorously? 

Reviewer #2: (No Response)

4. Have the authors made all data underlying the findings in their manuscript fully available?

Reviewer #2: (No Response)

5. Is the manuscript presented in an intelligible fashion and written in standard English?

Reviewer #2: (No Response)

6. Review Comments to the Author

Reviewer #2: (No Response)

7. PLOS authors have the option to publish the peer review history of their article (what does this mean? ). If published, this will include your full peer review and any attached files.

**Do you want your identity to be public for this peer review?** For information about this choice, including consent withdrawal, please see our Privacy Policy .

Reviewer #2: No

---

## [Editor Report · Acceptance letter]

PONE-D-24-56332R1

PLOS ONE

Dear Dr. Lv,

I'm pleased to inform you that your manuscript has been deemed suitable for publication in PLOS ONE. Congratulations! Your manuscript is now being handed over to our production team.

Kind regards,

on behalf of

Dr. Guojin Qin

Academic Editor

PLOS ONE